

# Assessments of the north hemisphere snow cover response to 1.5 °C and 2.0 °C warming

Aihui Wang, Lianlian Xu, and Xianghui Kong

Nansen-Zhu International Research Centre, Institute of Atmospheric Physics, Chinese Academy of Sciences, Beijing, 100029, China
*Correspondence to*: Aihui Wang (wangaihui@mail.iap.ac.cn)

**Abstract** The 2015 Paris Agreement has initialed a goal to pursue the global-mean temperature below 1.5 °C, and well below 2 °C above pre-industrial levels. As an important surface hydrology variable, the response of snow under different warming levels has not been well investigated. The community earth system model (CESM) project towards 1.5 °C and 2 °C warming targets, combined with CESM large Ensemble project (CESM-LE) brings an opportunity to address this issue. This study provides a comprehensive assessment of snow cover fraction (SCF) and snow area extent (SAE), and the associated Land Surface Air Temperature (LSAT) over North Hemisphere (NH) based on CESM-LE, CESM 1.5 °C and 2 °C projects, as well as CMIP5 historical, RCP2.6 and RCP4.5 products. Results show that the spatiotemporal variations of those modeled products are grossly consistent with the observation. The projected SAE magnitude change in RCP2.6 is comparable to that in 1.5 °C, but lower than that in 2 °C. The snow cover differences between 1.5 °C and 2 °C are prominent during the second half of 21$^{st}$ century. Changes in the LSAT and snow cover for 2071-2100 with respect to 1971-2000 exhibit the inconsistently spatial patterns. The contribution of increase in LSAT on the reduction of snow cover differs across seasons with the greatest in boreal autumn (49-55%) and the lowest in boreal summer (10-16%). The snow cover uncertainties induced by the ensemble variability show time invariant across CESM members, but increase with the warming signal among CMIP5 models. This feature reveals that the model physical parameterization plays a predominant role on the long-term snow simulations, while they are less affect by the climate internal variability.

## 1. Introduction

Snow mass over ground is one of the important surface hydrology elements. Due to the unique physical properties, such



as high albedo, emissivity and absorptivity, low thermal conductivity, and roughness length, snow strongly affects the

exchange in energy and water between land and atmosphere over cold regions (Robinson and Kukla, 1985; Zhang, 2005).

The snowpack is a moisture reservoir, and it stores rainfall (or snowfall) in winter and recharges the surface runoff and

ground water in spring (Zakharova et al., 2011; Belmecheri et al., 2016), and it is also an insulator for heat and radiation

which blocks the solar radiation arriving at the soil surface, as well as protects the heat loss from ground to atmosphere in

winter time. At the snow covered areas over high latitude, the ground temperature is usually higher than the air temperature

(Stieglitz et al., 2003). Furthermore, snow on the ground influences the rainfall in remote regions through the large-scale

atmospheric circulations (Liu and Yanai, 2002; Souma and Wang, 2010; Peings et al., 2013), and it was extensively used in

the data assimilation to improve the climate prediction skill (e.g., Dawson et al. 2016).

Snow ablation and accumulation are determined by many factors such as the land surface air temperature (LSAT) and

surface radiation. In general, increase in LSAT would enhance the ratio of rainfall to total precipitation over land as well as

speed up the snow melting, as a result, shorten the snow retention time on the ground (Smith et al., 2004). During past three

decades, observation evidences have shown that the annual snow area extent over the Northern Hemisphere (NH) have

reduced substantially ( Brown, 2000; Dye, 2002), and such terrestrial changes partially attribute to the increase in air

temperature (Mccabe and Wolock, 2010). Based on the 5th Coupled Model Intercomparison Project (CMIP5) (Taylor et al.,

2012), researchers have found that the projected surface warming would lead to earlier snowmelt (Oki and Kanae, 2006)

with a low rate in 21$^{st}$ century (Musselman et al., 2017). The relationship between snow cover and surface air temperature

has been discussed in many literatures (Cohen and Entekhabi, 1999; Laternser and Schneebeli, 2003; Mote, 2003). However,

comprehensive assessments of the snow cover response to different warming levels (e.g., 1.5 °C and 2.0 °C above

pre-industrial levels, hereafter referred as to 1.5 °C and 2.0 °C for short) have not been extensively performed.

The impacts of global warming on terrestrial variables have been investigated in various studies, and most of them have

focused on the risks avoiding 2 °C warming (Meinshausen et al., 2009; May, 2012; Salzmann et al., 2012; Schaeffer and

Hare, 2012). Recently, science communities argued that 1.5 °C warming would significantly reduce climate risk as compared

to 2 °C warming, and the 2015 Paris agreement initialed a goal to pursue the Global Mean Air Temperature (GMAT) below

1.5 °C, and well below 2 °C above the pre-industrial levels (UNFCCC, 2015). The academic community has shown agreat



interest on this initiative ( Boucher et al., 2016; Hulme, 2016; Peters, 2016; Rogelj and Knutti, 2016; Schleussner et al., 2016; Mitchell et al., 2017). The Intergovernmental Panel on Climate Change (IPCC) has also scheduled to propose a special report on the impacts of 1.5 °C in 2018 (http://www.ipcc.ch/report/sr15/pdf/information_note_expert_review.pdf). However, present comparison studies regarding to the differences between 2 °C and 1.5 °C are all through analyzing CMIP5 outputs

under the Representative Concentration Pathway (RCP) scenarios ( Vuuren et al., 2011; Schaeffer and Hare, 2012; Schleussner et al., 2016). For example, based on the CMIP5 model outputs, Schleussner et al. (2016) assessed the impacts of 1.5 °C and 2 °C warming levels on the extreme weather events, water availability, agricultural yields, sea-level rise and risk of coral reef loss, and concluded that substantial risk reductions with 1.5 °C compared to 2 °C warming, and further showing the regional differentiation in both climate risks and vulnerabilities. Indeed, the 1.5 °C is a relatively low warming target to

achieve with regard to the projections in RCPs. Jiang et al. (2016) showed that the probability of 2 °C warming before 2100 is 26, 86, and 100% for the RCP2.6, RCP4.5 and RCP8.5 respectively crossing all available CMIP5 model outputs. From these premise, there should be much higher probability for 1.5 °C occurrence. The RCPs were not specifically designed for targeting the climate impacts and risks for different warming levels such as 1.5 °C and 2 °C. In RCPs, the projected surface air temperature rising and the greenhouse gas emission are near-linear relationship (IPCC, 2014), leading to some research

gaps for climate change (e.g., Boucher et al., 2016; Peters 2016).

       To reduce above gaps, the Community Earth System Model (CESM) research group at the National Center for Atmospheric Research (NCAR) has performed a set of ensemble modeling experiment under the emulated concentration pathway leading to the stable 1.5 °C and 2 °C warming targets by 2100 (Sanderson et al., 2017). These experiment is the first earth system model simulation project towards both 1.5 °C and 2 °C warming goals. Together with the CESM Large

Ensemble (CESM-LE), above simulations provide the best available datasets to assess the potential impacts and risks with regard to 1.5 °C and 2 °C warming on both climatic and environmental elements.

       Based on above CESM simulations, CMIP5 model outputs, as well as the observed snow cover fraction datasets, this study extensively investigates the spatiotemporal change in snow cover over NH for both historical (1920-2005) and future (2006-2100) periods at 1.5 °C and 2.0 °C warming levels, as well as under RCP2.6 and RCP4.5 scenarios. The reproductions

of CESM on snow cover are evaluated with both in-situ and satellite data. The contribution of LSAT change in the snow





cover will be also quantified. Furthermore, a prominent advantage is that above CESM ensemble simulations facilitate to take insight into the impacts of the internal climate variability on those surface variables, which will also be addressed in this study.

## 2.    Models, Scenarios, and Data

### 2.1  The CESM and snow cover

The CESM consists of the Community Atmosphere Model version 5, the land surface model (CLM4), parallel ocean program version 2, and the Los Alamos sea ice model version 4(Hurrell et al., 2013). The fully coupled CESM has been used in many studies and also adopted in CMIP5 project (Taylor et al., 2012). The CESM and its performance have been well reported in the special issue of the Journal of Climate (http://journals.ametsoc.org/topic/ccsm4-cesm1). The snow process in the CESM is described in the land component of CLM4, in which the snowpack on the ground is divided up to five layers according to snow depth. The life cycle of snow such as ageing, compaction, thawing, and sublimation, are parameterized, and the effects of black carbon, organic carbon, and mineral dust on snow are also considered in CLM4.0 (Oleson et al., 2010).

The SCF is defined as the fraction area of a land grid cell covered by snow. In the CESM, the SCF ($f_{sno}$) is described as (Oleson et al., 2010)

$$f_{sno} = \tanh \left\{ \frac{z_{sno}}{2.5Z_0 \left[ \min\left(\rho_{sno}, 800\right) / \rho_{new} \right]^m} \right\} \qquad (1)$$

where $Z_{sno}$ is the snow depth over the ground, m = 1 is a parameter, $Z_0$= 0.01 m is the momentum roughness length for soil, $\rho_{new}$ = 100 kg m$^{-3}$ is the density of new snow, and $\rho_{sno}$ is the density of current snow, computed as the ratio of snow water equivalent and $Z_{sno}$. Equation (1) is modified based on the satellite and in-situ observation data (Niu and Yang, 2007). In the CLM4.0, the SCF directly affects the surface hydrology and heat processes (Oleson et al., 2010). The snow products in offline CLM4.0 simulation have been well evaluated by both satellite and in-suit observation, and the general conclusion is that the model simulations have overall captured the temporal variations on both SCF and snow water equivalence, whereas the model presents too early but fast snow ablation process (Swenson and Lawrence, 2012; Toure et al., 2016; Wang et al.,



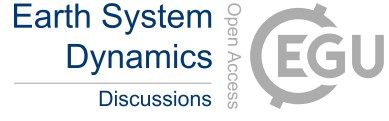

2016; Rhoades et al., 2017).

## 2.2 The CESM-LE project

The CESM-LE is a 40-member ensemble project, employing the fully coupled CESM version 1.1. Under the CMIP5

design protocol, all ensemble simulations have the same specified historical external forcing for 1920-2005 and future

scenario with RCP8.5 for 2006-2100, respectively. The ensemble member No.1 was run continuously from 1850 to 2100,

while the ensemble members No. 2 to 40 were restarted on January 1920 using the ensemble No.1 generated-initial condition

with slightly perturbations in air temperature (Kay et al., 2015). The horizontal resolution of the CESM-LE products is 0.9°

× 1.25°. Those products have been used on various studies such as investigating the impacts of the internal climate

variability on global air temperature variations (Dai et al., 2015), and the meteorological drought in China (Wang and Zeng,

2017). In this study, the monthly SCF and LSAT from the CESM-LE for 1920-2005 are treated as the historical simulations,

and the simulations from the member No. 1 for 1850-1919 are regards as the pre-industrial periods.

## 2.3   CESM 1.5°C and 2.0°C projects

The CESM 1.5 °C and 2.0 °C projects are specifically design for achieving the goal of the Paris Agreement of 2015

(Sanderson et al., 2017). The scenarios employ an emulator to simulate both the GMAT and emission concentration

evolution in earth system, and then the parameters in emulator were calibrated by the CESM simulations (Sanderson et al.,

2017). Based on the methodology established in Sanderson et al. (2016), three idealized emission pathways, including 1.5 °C

never-exceed (hereafter referred as to 1.5 °C), 1.5 °C overshoot (1.5 degOS), 2.0 °C never-exceed (hereafter referred as to

2.0 °C), were defined to limit the GMAT increasing within 1.5 °C and 2.0 °C by 2100 (Sanderson et al., 2017). In those

pathways, before 2017 the carbon emission follows RCP8.5, then the combined fossil fuel and land carbon emissions rapidly

decline to net-zero; finally, the emission fluxes are reduced even to negative which ensures the GMAT to achieve 1.5 °C and

2.0 °C warming targets by 2100. The difference between 1.5 degOS and 1.5 °C is that the carbon emission declines have

different evolutions. The detail of emulator establishment and scenarios design were described in Sanderson et al. (2017).

To "provide comprehensive resource for studying climate change in the presence of internal climate variability", a set of



multi-member projected simulations have been produced under three new scenarios, branching from the corresponding

historical simulations of CESM-LE in 2006 (Kay et al., 2015; Sanderson et al., 2017). There are 11 simulations (visited in

May 2017) available for both 1.5 °C and 2.0 °C scenarios, and the products can be downloaded from the earth system grid

website

(https://www.earthsystemgrid.org/dataset/ucar.cgd.ccsm4.lowwarming/). In the study, the monthly SCF and LSAT from

above ensemble simulations at 1.5 °C and 2.0 °C are analyzed.

## 2.4  CMIP5 data

The monthly SCF and LSAT from 12 models in CMIP5 for both the historical simulations (1850-2005) and future

projections (2006-2100) under RCP2.6 and RCP4.5 are used in this study (Taylor et al., 2012). The selection of models is

according to the data availability and the spatial resolution of each product, and only the first ensemble run (i.e. r1i1p1) in

each model is used. The models used in this study are BCC-CSM1.1, BNU-ESM, CanESM2, CCSM4, CNRM-CM5,

FGOALS-g2, FIO-ESM, GISS-E2-H, MIROC-ESM, MPI-ESM-MR, MRI-CGCM3, and NorESM1-M. The general

information of those models is summarized in Table S1. Those modeled SCF has been evaluated with the satellite

observation, and the results indicated that the model products overall were able to capture the spatial patterns, seasonal

change, and annual variation, but also showed the apparent disparities among different models ( Zhu and Dong, 2013; Xia

and Wang, 2015). The simulations from both RCP2.6 and RCP4.5 scenarios are chosen because the surface warming rates in

these two scenarios could be comparable to the 2.0 °C warming target to some extent (IPCC, 2013; Jiang et al., 2016). The

general information of 12 models can also be seen in Xia and Wang (2015). To facilitate the comparison, the monthly SCF

from 12 models are rescaled to 0.9° × 1.25° to match the resolution of CESM outputs.

## 2.5  Validation data

To validate the simulated SCF, the 0.05° MODIS Climate-Modeling Grid (CMG) version 6 daily snow cover products

are used (Hall and Riggs, 2016). It is well known that the satellite-based SCF has obvious biases when cloud presents. To

reduce the impacts of cloud cover, the daily confidence index (CI), defined as the percentage of clear-sky within a grid cell



from CMG product is applied to filter the CMG SCF products. Similar as the method used in Toure et al. (2016), we firstly

filter out the daily SCF data with CI less than 20 %, and then the daily filtered CMG SCF is averaged to monthly, and finally

they are aggregated in line with CSME-LE resolution (i.e., 0.9° × 1.25°).

Besides of MODIS SCF product, the monthly snow area extent (SAE) time series from the National Oceanic and

Atmospheric Administration Climate Data Record (NOAA-CDR) are also adopted to compare with the modeled products

(Robinson et al., 2012). The NOAA-CDR SAE is computed from the gridded monthly snow cover databases, deriving from

the NOAA weekly snow charts for 1966-1999 (Robinson, 1993) and the Interactive Multi-Sensor (IMS) daily snow product

for 1999 afterwards (Ramsay, 1998;Helfrich et al., 2007). The NOAA CDR SAE monthly time series averaged over NH are

obtained from the Rutgers University (http://climate.rutgers.edu/snowcover/).

### 3.   Methods

In this study, the pre-industrial periods are taken as the 1850-1919 mean in each modeled product, consistent with that

used in Sanderson et al. (2017). The SAE for each modeled product is computed as that the SCF multiplied the land area of

each grid cell. The monthly SAE and LSAT averaged over NH land area for 1920-2100 are then derived from all products.

The annual anomaly of each variable with its corresponding 1850-1919 mean denotes the change with respect to the

pre-industrial periods. The linear trend is derived from the least-square-fit method. The period of 1971-2000 is used as the

common baseline period. To qualify the change in the future, the mean value of each product for 2071-2100 is compared

with the baseline period. The standard derivation (STD) across CESM ensemble members or CMIP5 multi models represents

the spread of simulations due to ensemble variability. To address the contribution of change in SCF due to LSAT warming,

both the pattern correlation coefficient and the coefficient of determination between them for different seasons and different

products are also computed.

### 4.   Validation of modeled SCF

Figure 1 shows the mean (2001-2005) SCF from MODIS, CESM-LE ensemble mean, and CMIP5 ensemble mean. The

SCF differences of two ensemble means from MODIS and the STDs of their differences are also plotted. Overall, the spatial



patterns from three products are similar, with the greatest over the high latitudes and then reduce at the middle and low latitudes. The annual mean SCF averaged over entire NH land area is 17.97% for MODIS, 22.3 ± 0.26% (STD) for CESM-LE, and 16.24 ± 7.87% (STD) for CMIP5 for 2001-2005. Compared with the MODIS, the CESM-LE ensemble mean overestimates the SCF over most land areas with an exception at small portion in west Eurasia (Fig. 1d), while the CMIP5

ensemble mean is comparable to that in MODIS with slight underestimation over Eurasian continent, North America, and Greenland (Fig. 1e). Toure et al. (2016) evaluated the MODIS SCF with offline CLM4.0 simulations driven by the observation-based atmospheric forcing data set, and they found that the model overall underestimated the mean SCF average, in particular during melting season. They attributed the modeled SCF biases to the snow process parameterization, sub-grid effect in CLM4.0, as well as the forest coverage and cloud cover induced uncertainties in MODIS. Those issues still exist in

the CESM-LE. The overestimation SCF in CESM-LE in contrast with the underestimation by offline CLM4.0 is partially attributed to the biases in surface atmospheric forcing variables (e.g., precipitation, air temperature, humidity etc.), which are produced by the atmospheric model in CESM-LE, and the biases due to rainfall and snowfall separation are also responsible for the above SCF biases in CESM-LE (Wang et al., 2016).

The STDs of SCF differences from CESM-LE is generally less than 5% with the greatest locating at the western United

States, part of Eurasian middle latitude continent and the Tibetan Plateau (Fig. 1f). In contrast, the STDs from CMIP5 are above 10% over majority snow regions (Fig. 1g), which are greatly larger than the magnitude of their ensemble mean differences (Fig. 1e). In fact, the spread from CESM-LE is induced by the internal climate variability due to the interaction of intrinsic dynamical processes within the earth system, in which the slight perturbation of the initial condition in CESM-LE experiment would lead to different climate variability (Kay et al., 2015). The STD from CMIP5 is mainly caused

by the model structure and physical parameterization, in particular, the representation of snow process in different models because all models used the same external forcing (Taylor et al., 2012). Therefore, these results indicate that the SCF heavily relays on the physical process representation in the model, while the internal climate variability might play a relatively minor role.

Figure 2 shows the 12-month moving averaged SAE anomalies over NH from NOAA-CDR, CMIP5, and CESM-LE

ensemble mean during the period of 1967-2005. The full spread of CMIP5 12 models and CESM-LE 40 ensemble members



are also shown. The SAE from NOAA-CDR exhibits apparently annual variations with the anomaly varying within $\pm 2 \times 10^6$ km$^2$, while SAE from both CMIP5 and CESM-LE ensemble mean show less temporal variations. The spreads from both products are remarkable and their envelops cover most NOAA-CDR curves, implying that SAE from both modeled products are reasonable.

To further investigate the SAE temporal variations, we compute the R between modeled products and NOAA-CDR, and the linear trends of three products for the period of 1976-2005 (Table 1). For CESM-LE, the R varies between -0.41 and 0.55 with the mean 0.18 ± 0.17 (STD), and there are 35 of 40 members with the positive R, while for CMIP5 the R varies from 0.10 to 0.50 with mean 0.24 ± 0.12 (STD). The linear trends of SAE from all three products exhibit the reduction with the values being -3.98 × 10$^4$ km$^2$/year, -2.36 ± 0.76(STD) × 10$^4$ km$^2$/year, and -2.62 ± 1.33 ×10$^4$ km$^2$/ year for NOAA-CDR,

CSEM-LE, and CMIP5, respectively. The trend spreads from -4.35 × 10$^4$ km$^2$/year to -0.22 × 10$^4$ km$^2$/year across CESM-LE ensemble members, and from -5.22 × 10$^4$ km$^2$/year to -1.02 × 10$^4$ km$^2$/year for CMIP5 models, respectively. The ensemble mean of both modeled products underestimates the magnitude of SAE reduction. However, accounting for the model spreads in two products, both modeled SAE reductions are roughly comparable to that in NOAA-CDR. On the other hand, the majority members with positive R and the consistency in the reduction of SAE imply that both CMIP5 and CESM-LE

products can be used to represent the variations of SAE. It should be noted that the deficiencies of climate model in reproduction of snow are beyond this work, therefore there are not discussed in this study.

## 5.    Impacts of the LSAT on snow cover

### 5.1  Long-term SAE variations

To qualify the long-term SAE variations, Figure 3 shows the annual anomalies of both SAE and LSAT average over NH land area for the period of 1920-2100. All anomalies are respected to the mean value for the pre-industrial period. Both the ensemble mean and full spread of ensemble members are displayed. There are distinctly temporal variations in the longterm evolution and the magnitude diversity among different products from both SAE (Fig.3a) and LSAT (Fig.3b). During the period of 1920-2005, the ensemble SAE anomaly from both CESM-LE and CMIP5 shows similar annual variations with the

correlation coefficient 0.86, but the actually values from CESM-LE are consistently larger than that from CMIP5. Before



early 1960s, the time variability of SAE is relatively small, and afterwards it shows apparently decreasing tendency. Overall, SAE reduction from CMIP5 ensemble is much larger than that from CESM-LE ensemble mean. For example, from 1920 to 2005, the annual SAE ensemble mean reduces about $0.75 \times 10^6$ km$^2$ from CESM-LE, while this value is $1.32 \times 10^6$ km$^2$ for CMIP5. For the future period, during 2005-2050 the linear trends of SAE are all negative, varying between $-4.92 \times 10^4$ km$^2$

/year (2.0 °C) and $-2.37 \times 10^4$ km$^2$/year (RCP 2.6), while after 2050 the trends from both RCP 2.6 ($0.32 \times 10^4$ km$^2$/year) and 1.5 °C ($0.26 \times 10^4$ km$^2$/year) turn to positive. Moreover, before 2050 the ensemble mean SAE anomaly from CMIP5 is below those from CESM-based simulations, but after-2050 they are comparable to each other from both RCP 2.6 and 1.5 °C. Nevertheless, although the ensemble mean SAE shows overall downtrend for future period, the upper envelop of the spread from RCP 2.6 gives positive SAE anomalies in a few years, with the maximum value about $1.0 \times 10^6$ km$^2$. This feature

implies that the projected SAE under RCP2.6 in some models would be above the pre-industrial level.

In contrast, the LSAT anomaly exhibits the continuously increasing tendency (Fig. 3b). The linear trends of LSAT from ensemble mean are 0.022, 0.026, 0.034, and 0.043 °C/year for RCP 2.6, 1.5° C, RCP 4.5, and 2.0 °C for 2006-2050, respectively. Similar as the SAE variations, since 2050 the LSAT trends turns to negative in both RCP2.6 (-0.03 °C/decade) and 1.5 °C (-0.02 °C/decade), and the magnitudes of trends from both RCP 4.5 and 2.0 °C also become smaller compared

with those in early period. Furthermore, Fig.3 also shows that the spread of the ensemble members displays different variability for different products.

To examine the evolution of SAE anomaly spread among different ensemble members, we have computed the STDs of SAE across all members in each year and shown in Fig. 4. The STDs from both CESM and CMIP5 show apparently annual variations. From the entire period of 1920-2100, the annual STDs from CESM changes slightly with time, varying between

$0.3 \times 10^6$ km$^2$ and $0.7 \times 10^6$ km$^2$, while the STDs from CMIP5 increases with time distinctly, with the increase in magnitude up to $1.4 \times 10^6$ km$^2$. Similar results can also be derived from LSAT (Fig. S1). This suggests the uncertainty induced by the climate internal variability is an inherent property in the climate system and it is almost stationary, while their uncertainty from CMIP5 multi model simulations increases with warming signals. Therefore, caution should be taken when the CMIP5 outputs from multi model ensemble are used to address the long term change of surface variables.



## 5.2 Future SCF and LSAT changes in both 1.5 °C and 2.0 °C

Figure 5 shows the 30-year annual mean SCF differences between 2071-2100 and 1971-2000 from both 1.5 °C and 2.0 °C scenarios, respectively. Both products show consensus change of SCF for 2071-2100, and the NH average SAE change is -1.69 ×$10^6$ km$^2$ in 1.5 °C, and -2.36×$10^6$ km$^2$ in 2.0 °C. The annual mean ensemble differences are not uniformly distributed. The largest magnitude change could be above 10%, appearing at the mountain ranges in the middle latitudes, such as the Iran Plateau, north Canada, west America along the Rocket Mountain, and west Tibetan Plateau. In comparison of the ensemble mean SCF between 1.5 °C and 2.0 °C for 2071-2100 (Fig. 5c), the differences are generally below 4% over majority snow areas (and the SAE difference is $0.67 \times 10^6$ km$^2$) with the largest difference at the same locations as the largest SCF reduction with regard to pre-industrial levels (Figs. 5a and 5b). In contrast, the ensemble mean LSAT exhibits the largest warming over polar region in the future and the warming magnitude reduces over middle and low latitudes (Figs.5d and 5e). The increase in LSAT for 2071-2100 exceeds 4 °C along the coastline of the Arctic Ocean. The prominent warming over polar region represents the polar amplification effect, which might be related to the sea ice change (Screen and Simmonds, 2010). The inconsistent spatial variations of LAST and SCF suggest that the LSAT is not the only factor to determine the SCF change. The contribution of LSAT warming on the SCF reduction would be further addressed later.

Figure 6 illustrates the signal-to-ratio (referred as to SNR), defined as the ratio of the ensemble mean change to the STDs of change across the ensemble members. This metric represents the relative importance of external forcing and the climate internal variability on the variable change (Kay et al., 2015;Wang and Zeng, 2017). The SNR exceeds 2 over majority areas for both variables (Fig. 6). This feature implies that both LSAT and SCF changes are dominantly affected by the external (or anthropogenic) forcing, with slightly triggered by the climate internal climate variability. The spatial pattern of SNR for both SCF and LSAT are broadly consistent with each other over snow regions, but their magnitude for SCF is much smaller than that for LSAT. Figure 6 also shows that the SNR in 2 °C is generally larger than that from 1.5 °C. This further reflects that with increase in warming levels, the external forcing plays more predominant role on the change in LSAT and SCF. This result is consistent with the changes in the meteorological drought in Wang and Zeng (2017).

## 5.3 Contribution of LSAT on snow cover reduction



Under the climate change background, the increasing in surface air temperature in recent decades is one of the most prominent features. In the CESM, the surface air temperature with a 0 °C threshold is used to separate the rainfall and snowfall. Therefore, the increase in surface air temperature would reduce the chance of snowfall, but enhance the rainfall occurrence. An outstanding question is: to what degree the increase in local LSAT is responsible for the snow cover reduction by 2100?

To address above question, we compute the pattern-correlation (R) between SCF and LAST change for 2071-2100 versus 1971-2100 over NH from 1.5 °C, 2 °C, RCP 2.6, and RCP 4.5 scenarios (Fig. 7a). The R is separately derived from both annual and four seasons, i.e., boreal winter (December-February, DJF), spring (March-May, MAM), summer (June-August, JJA), and autumn (September-November, SON). As discussed previous, the increase in LSAT will reduce the local snow fraction, therefore it is undoubtedly that R should be negative. For all four seasons and annual R, the ensemble mean R is smaller than -0.3 with all passing the significant test at the 95% confidence level. The magnitude of R shows clearly seasonal variations, with the highest in boreal autumn and lowest in boreal summer. For example, R varies from -0.70 (RCP 2.6, RCP 4.5) to -0.75(1.5 °C and 2.0 °C) in boreal autumn, and from -0.30 (RCP 4.5) to -0.40 (1.5 °C) in boreal summer. Furthermore, we see clearly that the ensemble variability (denoted as STDs of R in Fig. 7a) from CESM-based products are relatively small when it is compared to the ensemble mean R, as well as compared to those from CMIP5. This further illustrates that external forcing is the predominant factor for above relationship.

To quantify of the contribution of LSAT warming on snow reduction, we adopt an index, correlation of coefficient (CD), defined as the percentage of squared pattern correlation (CD = $100\% \bullet R^2$) for four seasons and annually in different scenarios (Fig. 7b). The CD denotes the percentage of SCF reduction explained by the LSAT increase. Similar as R, the CD shows clearly seasonal variations with the greatest in boreal autumn (49% ~ 55%) and lowest in summer (10% ~16%), and the STDs of CD are also larger in CMIP5 than in CESM-based simulations. Although the CDs from CMIP5 ensemble mean are slightly smaller than that from CESM-based simulations, overall the two from the specific season are comparable. For example, the CDs of ensemble annual mean difference are about 50% for all products. This means that the LSAT change could explain approximately half of SCF reduction annually, and the change in SCF would also be affected by other factors.



## 6. Conclusions

This study investigates the long-term change in the SCF and SAE associated with LSAT over NH during the period of

1920-2100. We have analyzed the simulations from CESM-LE, CESM 1.5 °C and 2.0 °C projects (Sanderson et al., 2017),

as well as simulations from historical, RCP2.6 and RCP4.5 from CMIP5 12 models (Taylor et al., 2012). The model

simulated snow cover products are evaluated with the MODIS and NOAA-CDR observation. We emphasize on the

responses of both SCF and SAE under different warming levels. The reduction of snow cover due to increase in LSAT is

quantified. The relative importance of climate internal variability and external forcing on the change in both SCF and LSAT

and their relationship are also addressed.

We find that the ensemble annual mean SCF from both CMIP5 and CESM-LE simulations can broadly capture the

spatial pattern of MODIS, with slightly underestimation in CMIP5, but overestimation in CESM-LE. Annual SAE from

ensemble mean of CMIP5 and CESM-LE, and NOAA-CDR all display significant reduction trends for the period of

1967-2005. Compared to the pre-industrial periods, the SAE anomalies from CMIP5 and CESM simulations show gross

similarities in the annual variations. Overall, the annual ensemble SAE displays downtrend, but the LSAT exhibits uptrend

for the long-term period of 1920-2100. However, the actual variability differs in different products for different time periods.

The trends of projected SAE (LSAT) from all products are negative (positive) for the period of 2006-2100, but they become

positive (negative) during the second half of 21$^{st}$ century in both RCP 2.6 and 1.5 °C. The magnitude of SAE anomaly in

RCP 2.6 is comparable to that in 1.5 °C, while it is smaller than that in 2.0 °C. Furthermore, the STDs of SAE induced by

ensemble variability show time invariant across CESM ensemble members, but increases with warming signal among

CMIP5 models. Therefore, cautions should be taken when the multi models projected surface variables are analyzed.

For 30-year mean change between 2071-2100 and 1971-2000, the variable (SCF or LSAT) show similarly consensus

spatial variations in both 1.5 °C and 2 °C. However, the spatial patterns of SCF change are not consistent with those of LSAT

change. That is, regions with the largest SCF differences do not show the greatest LSAT warming. In comparison with the

ensemble mean SCF between 1.5 °C and 2 °C for 2071-2100, the SCF differences are less than 4% over most snow grid cells

and SAE difference is $0.67 \times 10^6$ km$^2$. Moreover, we also find that the external forcing plays the predominant role on the

changes in future with respect to the base period in both SFC and LSAT, with the SNR of differences over 2 in general.



The significant negative R between projected LSAT and SCF change for 2071-2100 versus the baseline period of 1971-2000 denotes that the SCF reduction strongly relies on the LSAT warming. Through analyzing the CDs, we find that more than 50% of boreal autumn and annual reduction of projected SCF over NH is attributed to the increase in LSAT, whereas this value is less than 16% in boreal summer. Furthermore, STDs of CDs are much larger from CMIP5 than in CESM-based simulations. This feature implies that the SAE uncertainties mainly come from the physical structure and the representation of snow process in different CMIP5 models, while they are less affected by the climate internal variability from CESM ensemble.

Finally, we provide a comprehensively analysis of both SCF and SAE from the CESM and CMIP5 simulations for both historical and future periods in different warming or emission scenarios. Under different scenarios (e.g., RCP2.6, RCP4.5, 2 °C, and 1.5 °C warming above pre-industrial levels), the snow cover response to LSAT warming varies with season and differs in products. In conclusion, surface warming is partially responsible for the surface snow change. Researches have been shown that the Arctic sea ice has greatly impacted on the snow cover (Kapnick and Hall, 2012), and the human activities induce the black carbon reducing the snow surface albedo and enhance the solar radiation absorbed by the snow, as a result, acceleration of snow reduction (Flanner et al., 2007). Those factors on snow reduction are beyond this topic. More works would be carried out in the future.

**Acknowledgments**

The work was supported by the Key project of Ministry of Science and Technology of China (2016YFA0602401). The authors would like acknowledge the Community Earth System Model (CESM) modeling group for making their datasets available through https://www.earthsystemgrid.org/. We acknowledge the World Climate Research Program (WCRP) Working Group on Coupled Modelling, which is responsible for CMIP5, and the data sets were downloaded from http://cmip-pcmdi.llnl.gov/cmip5/data_portal.html



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



# Tables

Table 1: Correlation coefficient (R) of snow area extent (SAE) between CESM-LE and NOAA-CDR, between CMIP5 and NOAA-CDR, and annually linear trend of SAE from above three products for the period of 1976-2005, respectively. The mean, standard deviation, maximum, and minimum of the corresponding statistics from CESM-LE multi-member and CMIP5 multi-model are also displayed. The value in the last column is the annually linear trend of SAE from NOAA-CDR. The values with superscript star denote the R or Trend passing 95% significant level test.

|  | R (CESM-LE, NOAA-CDR) | R (CMIP5, NOAA-CDR) | Trend CESM-LE $10^4$km$^2$/year | Trend CMIP5 $10^4$km$^2$/year | Trend NOAA-CDR $10^4$km$^2$/year |
|---|---|---|---|---|---|
| **Mean** | 0.18 | 0.24 | -2.36* | -2.62* | -3.98* |
| **Standard deviation** | 0.17 | 0.12 | 0.76 | 1.33 | --- |
| **Maximum** | 0.55* | 0.50* | -0.22 | -1.02 | --- |
| **Minimum** | -0.41* | 0.1 | -4.35* | -5.22* | --- |

**Figures:**

Figure 1 Averaged annual snow cover fraction over Northern Hemisphere land area from a) MODIS, b)
CESM-LE ensemble, c) CMIP5 ensemble, the difference between d) CESM-LE ensemble and MODIS,
e) CMIP5 ensemble and MODIS, f) and g) are the standard deviation of c) and d), respectively. The
average was taken for period of 2001-2005.





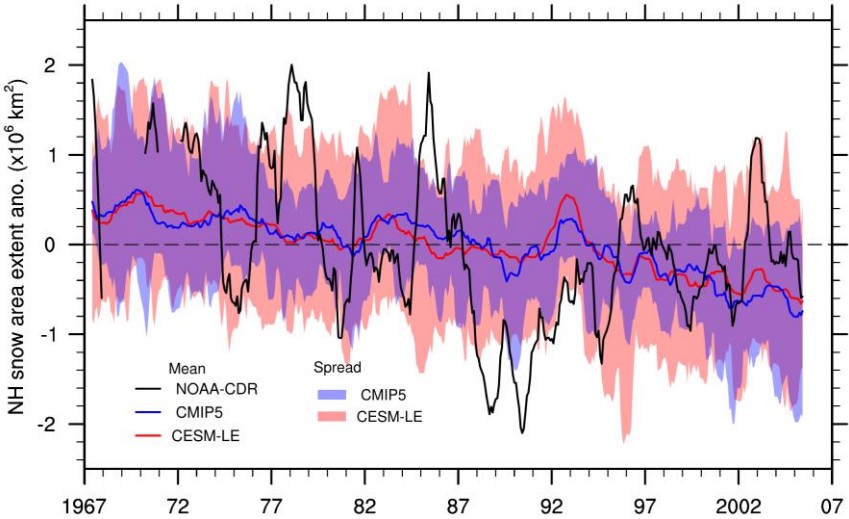

Figure 2 Time series of snow area extent (SAE) anomalies from NOAA-CDR, CMIP5 12 models, and CESM-LE 40 ensemble members over Northern Hemisphere land area for the period of 1967-2005. The spreads of CMIP5 12 models and from CESM-LE 40 ensemble members are shaded.



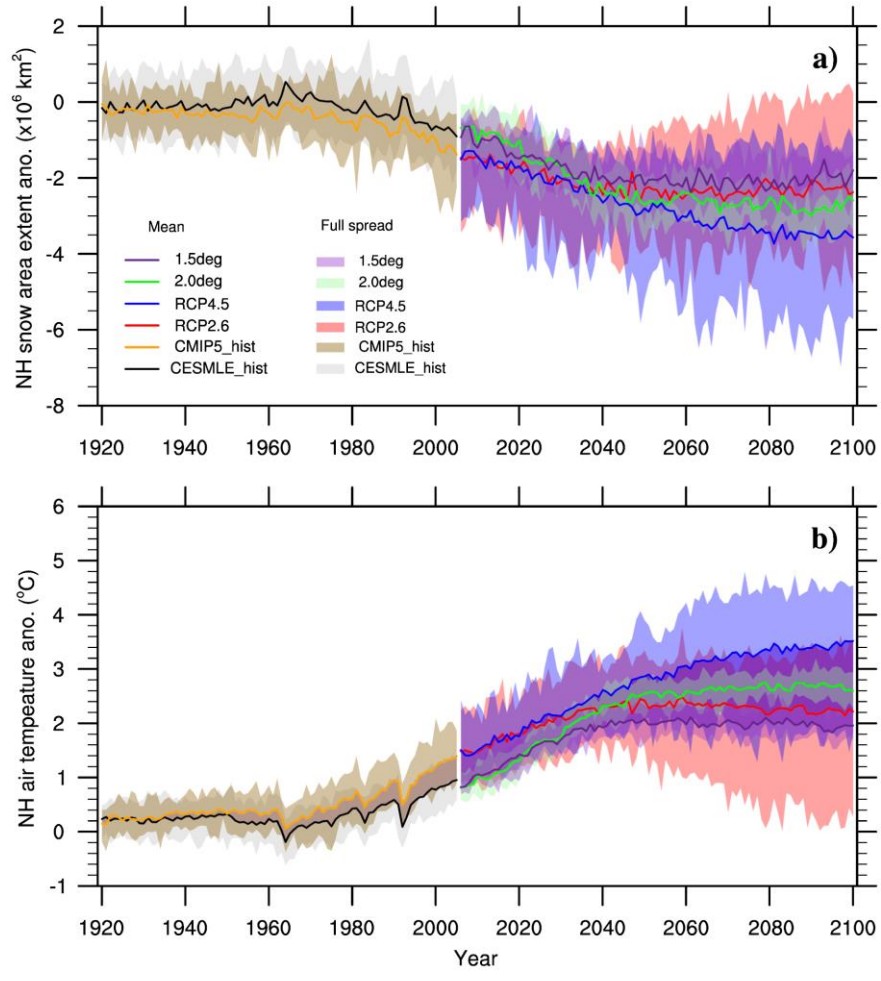

Figure 3 Annual time series of a) snow area extent (SAE) anomalies, and b) land surface air temperature (LSAT) anomalies over Northern Hemisphere for 1920-2100. The different colors represent the simulations from different projects with different scenarios. The shaded represents full spread from simulations in both CMIP5 models and CESM ensemble members. Note that "1.5 deg" and "2.0 deg" represent simulations from 1.5°C and 2.0°C scenarios, respectively.

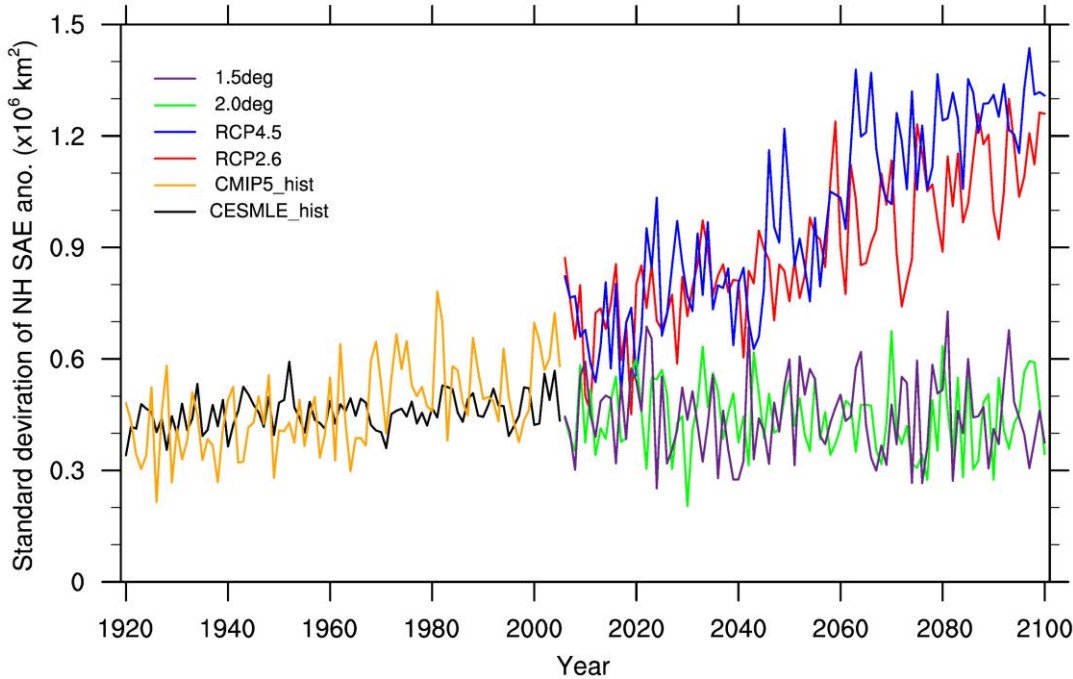

Figure 4 The annual standard deviation of snow area extent anomaly due to the ensemble variability for 1920-2100. Results from CESM-LE, CMIP5 historical, RCP2.6, RCP4.5, 1.5°C and 2.0°C scenarios are shown.



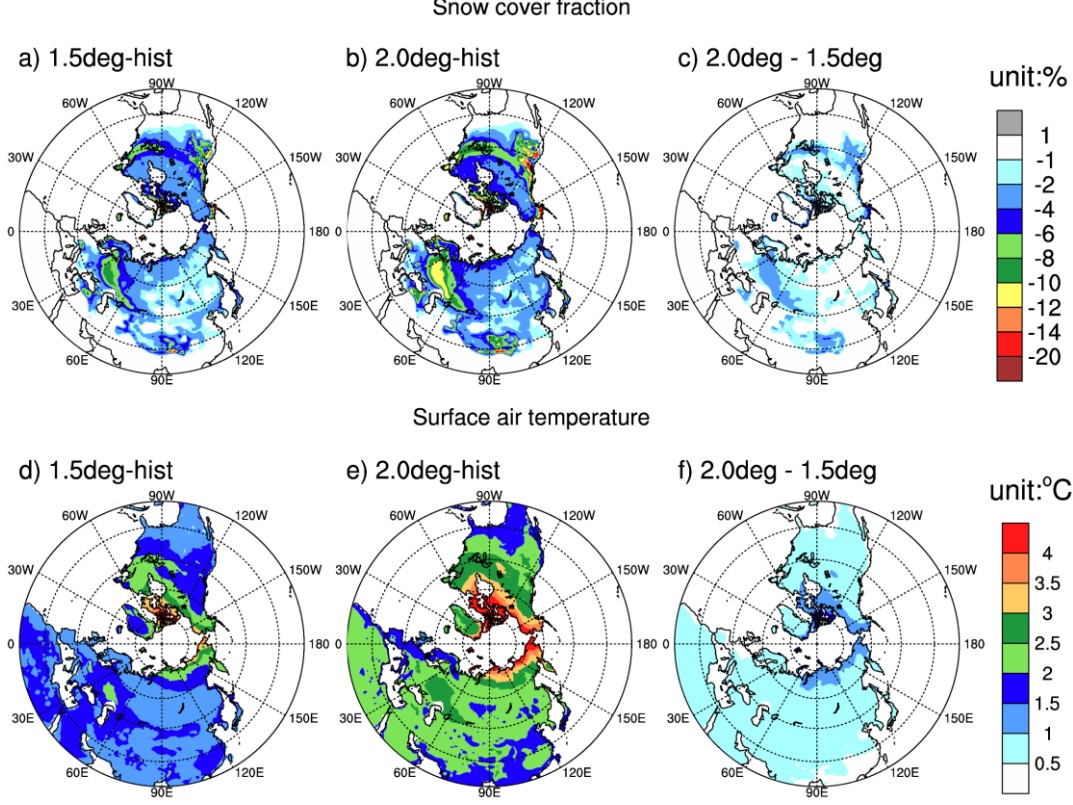

Figure 5 Snow cover fraction (top) and land surface air temperature (bottom) differences between 2071-2100 and 1971-2000 over Northern Hemisphere land area from a) 1.5deg, b) 2.0 deg, and c) 2.0deg minus 1.5deg. d)-f) are the land surface air temperature differences correspondingly for a)-c) respectively. "hist" is the ensemble mean for 1971-2000 from CESM-LE.





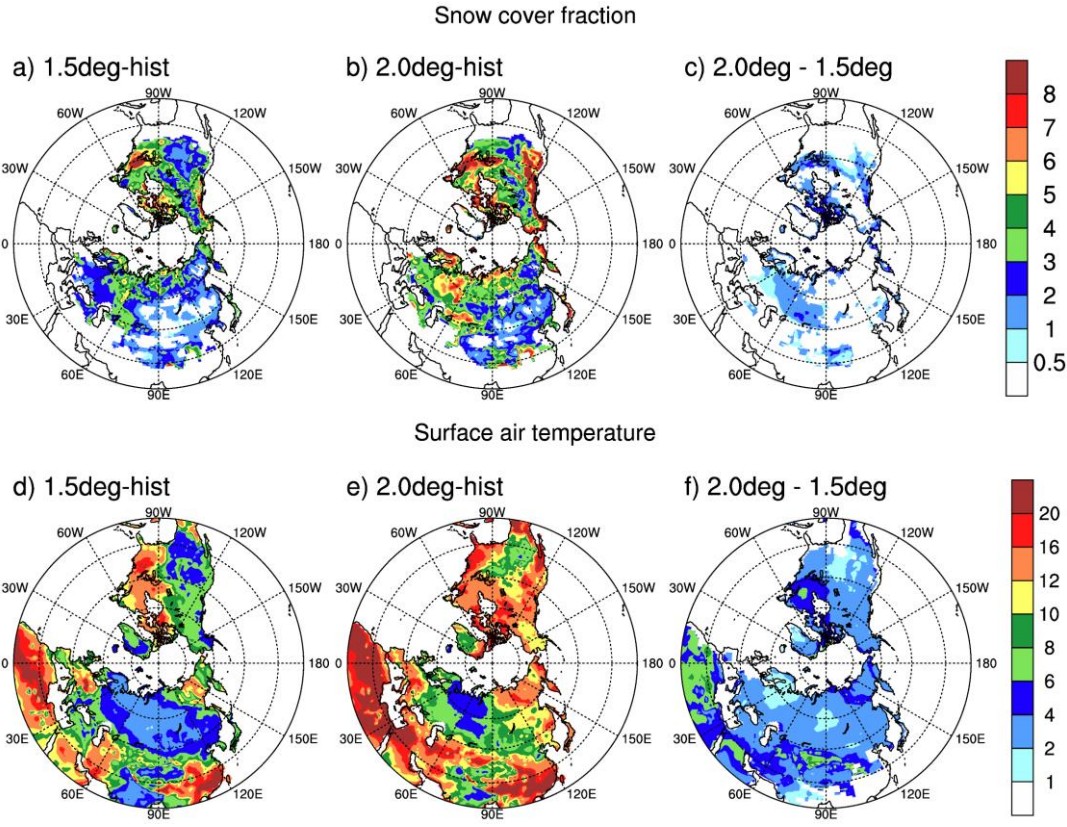

Figure 6 Similar as Figure 5, but for the signal-to-ratio (SNR) of snow cover fraction (a-c) and land surface air temperature differences (d-f) between 2071-2100 and 1971-2000 over Northern Hemisphere land area. The SNR was computed as the ratio of change in ensemble mean to the standard deviation due to the ensemble variability



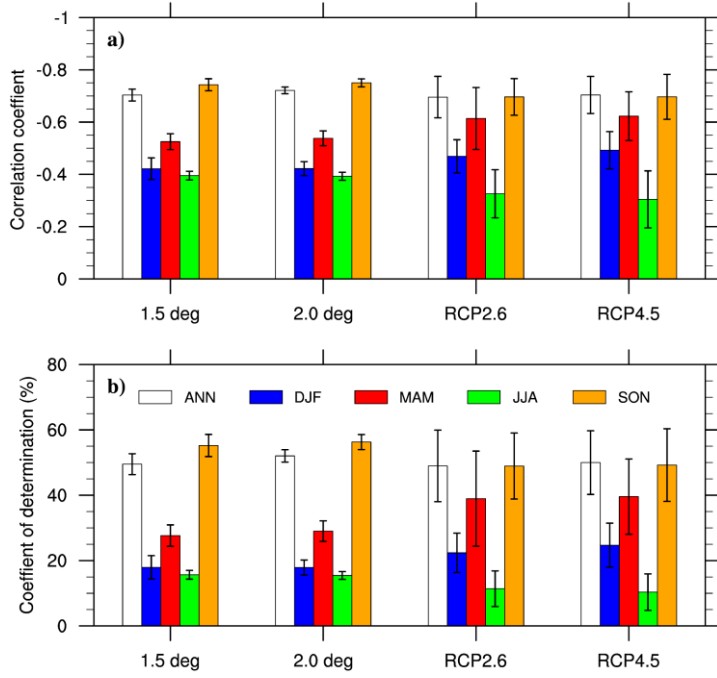

Figure 7 Pattern correlation a) and the coefficient of determination b) between surface air temperature change and snow cover fraction change from 1.5°C, 2.0°C, RCP2.6, RCP4.5 scenarios. The changes are computed as the difference between 2071-2100 and 1971-2000. The bar represents the ensemble mean, and the vertical line is the standard deviation from 12 models (RCP2.6 and RCP4.5) or 11 CESM simulations (1.5°C and 2.0°C).