# Peer review of "Assessments of the north hemisphere snow cover response to 1.5 °C and 2.0 °C warming"

_Earth System Dynamics, 2017_

## Referee Comment (RC1) · Anonymous Referee #1 · 19 Jan 2018

This study purports to examine the response of Northern Hemisphere snow cover in the Community Earth System Model 1 (CESM1) under 1.5°C and 2.0°C of global mean warming above preindustrial levels. I am unable to recommend further publication of this manuscript as it provides a very cursory examination of the question under discussion and in my opinion, does not reach substantial conclusions that have not been reported elsewhere. The methods used appear reasonable, however the scientific aims and results are poorly communicated throughout the paper. I provide some more specific criticisms below.

The rational for providing Figures 1 and 2 is unclear. Presumably it is to evaluate the simulation of climatological snow cover and its variability for the CMIP5 ensemble and the CESM1 model in particular. These evaluations have been performed elsewhere

[Figure]

for CMIP5 for a variety of seasons (for example, Li et al. evaluate CMIP5 ensemble mean SCF during DJF (Figure 2b), Thackeray et al. evaluate seasonal evolution of SAE in CMIP5, Brutel-Vuilmet et al. evaluate March/April SCF climatologies in CMIP5, and Rupp et al. evaluate springtime SAE variability from the CMIP5 ensemble). These results suggest that snow cover biases change over the course of the season and that annual mean SCF may have lower bias due to compensating differences throughout the year. Regarding CESM, as you note, the bias computed in this paper is in contrast to previous offline evaluation of CESM's component land model, CLM. While it is possible, or even probable that biases in CESM1 precipitation and air temperature are responsible for the differences in SCF, I don't believe the Wang et al. study, which is referenced in the text, examines this bias for CESM's atmospheric model. It would be helpful to confirm any air temperature or precipitation biases in CESM1 compared to the corrected MERRA output used by the Toure et al. study to examine CLM. For Figure 2, the range of correlations between SAE anomalies in the NOAA-CDR and the CESM-LE or CMIP5 ensembles are expected since the specific climate trajectories of the historical ensembles are not constrained by observations. Presumably the positive values obtained for the ensemble mean correlations with the NOAA-CDR result from negative trends in the datasets. Also note that NOAA-CRD has been shown to have erroneous trends in SAE during the fall and potentially during the spring as well (Brown and Derksen, Hori et al., Mudryk et al.); it is unclear to what extent this will affect trends in annual-mean SAE.

Figure 3: These results appear consistent with results in IPCC Fifth Assessment Report (e.g. Figure 12.32 in Chapter 12 of the Working Group I contribution). I don't believe the projections of annual mean SAE from the CESM historical ensemble or low warming ensembles are sufficiently unusual to require an update of these results. In particular the historical ensembles appear consistent with the differences in NH LSAT (colder and more snow cover in CESM-LE than CMIP5) and the projections of SAE for each of the four scenarios reflect the various levels of warming.
Figure 4: The increase in standard deviation of SAE anomalies reflects the spread in CMIP5 model SAE sensitivity (different models lose snow at different rates during the 20th and 21st centuries, probably mostly due to the spread in warming rates over the 21st century, although this latter point is not examined by the authors). For the single model ensembles (CESM-LE hist, 1.5deg and 2.0deg) this spread is not being sampled.

Figure 5 and 6: These are the first figures which deal with the topic of the paper as discussed in the abstract and introduction. It's unclear why CMIP5 models were examined in the preceding portion of the paper given that they are not used here (although I grant there are potential difficulties in comparing model output at a given warming level taken from transient RCP scenarios and partially equilibrated simulations such as the CESM low warming simulations). While the results presented in Figures 5 and 6 appear reasonable, I don't believe they are analyzed sufficiently or novel enough to form the basis of this paper.

"The spatial pattern of SNR for both SCF and LSAT are broadly consistent with each other over snow regions, but their magnitude for SCF is much smaller than that for LSAT." It's not clear to me which aspects of the SNR patterns are consistent between that of SCF and that of LSAT.

Figure 7: Figure 7b is not providing any additional information from 7a. To the level of detail presented, the results do not appear depend to depend on the forcing scenario. No interpretation is provided or discussed regarding the seasonal dependence.

Brown, R., and Derksen, C.: Is Eurasian October snow cover extent increasing? Environmental Research Letters, 8, 024006, doi:10.1088/1748-9326/8/2/024006, 2013.

Brutel-Vuilmet, C., Ménégoz, M., and Krinner, G.: An analysis of present and future seasonal Northern Hemisphere land snow cover simulated by CMIP5 coupled climate models, The Cryosphere, 7, 67-80, https://doi.org/10.5194/tc-7-67-2013, 2013.
Hori, M., Sugiura, K., Kobayashi, K., Aoki, T., Tanikawa, T., Kuchiki, K., Niwano, M., and Enomoto, H.: A 38-year (1978–2015) Northern Hemisphere daily snow cover extent product derived using consistent objective criteria from satellite-borne optical sensors. Remote Sens. Environ., 191, 402–418, Doi:10.1016/j.rse.2017.01.023, 2017.

Li, Y., T. Wang, Z. Zeng, S. Peng, X. Lian, and S. Piao (2016), Evaluating biases in simulated land surface albedo from CMIP5 global climate models, J. Geophys. Res. Atmos., 121, 6178–6190, doi:10.1002/2016JD024774.

Mudryk, L. R., P. J. Kushner, C. Derksen, and C. Thackeray (2017), Snow cover response to temperature in observational and climate model ensembles, Geophys. Res. Lett., 44, 919–926, doi:10.1002/2016GL071789.

Rupp, D.E., P.W. Mote, N.L. Bindoff, P.A. Stott, and D.A. Robinson, 2013: Detection and Attribution of Observed Changes in Northern Hemisphere Spring Snow Cover. J. Climate, 26, 6904–6914, https://doi.org/10.1175/JCLI-D-12-00563.1

Thackeray, C. W., C. G. Fletcher, and C. Derksen (2015), Quantifying the skill of CMIP5 models in simulating seasonal albedo and snow cover evolution. J. Geophys. Res. Atmos., 120, 5831–5849. doi: 10.1002/2015JD023325.

---

## Author Comment (AC1) · 22 Jan 2018

The authors thank for the reviewer#1 critical comments. According to the ESD interactive discussion process, we will first try to reply the reviewer#1' comments in this short text, and then we will revise our manuscript throughout. In the following text, we firsty copied the reviewer's comments, and then made one-by-one reply after "R:".

This study purports to examine the response of Northern Hemisphere snow cover in the Community Earth System Model 1 (CESM1) under 1.5_C and 2.0_C of global mean warming above preindustrial levels. I am unable to recommend further publication of this manuscript as it provides a very cursory examination of the question under discussion and in my opinion, does not reach substantial conclusions that have not been

reported elsewhere. The methods used appear reasonable, however the scientific aims and results are poorly communicated throughout the paper.

R: The snow simulations were essentially evaluated and reported in many literatures. We appreciated the reviewer# 1 listed a bunch of papers about snow in the models or in the observations. However, none of them discussed the snow cover change under different warming levels, say 1.5 and 2.0deg, which is one of our purposes of current study. Indeed, there are some works who used the CMIP5 outputs to examine the relationship between snow cover and surface air temperature. As we described in the introduction, the CMIP5 simulations under RCP scenarios were the model response from the Radiation Forcing at the top of the atmosphere, but not for specific surface air temperature warming levels. The CESM low warming project aimed to the specific (i.e., 1.5 and 2 deg) warming targets. Moreover, this manuscript has also discussed he impacts of the climate internal variability and external forcing on the response of snow cover on the different warming levers, which haven't been addressed and reported anywhere.

I provide some more specific criticisms below. The rational for providing Figures 1 and 2 is unclear. Presumably it is to evaluate the simulation of climatological snow cover and its variability for the CMIP5 ensemble and the CESM1 model in particular. These evaluations have been performed elsewhere C1 for CMIP5 for a variety of seasons (for example, Li et al. evaluate CMIP5 ensemble mean SCF during DJF (Figure 2b), Thackeray et al. evaluate seasonal evolution of SAE in CMIP5, Brutel-Vuilmet et al. evaluate March/April SCF climatologies in CMIP5, and Rupp et al. evaluate springtime SAE variability from the CMIP5 ensemble). These results suggest that snow cover biases change over the course of the season and that annual mean SCF may have lower bias due to compensating differences throughout the year. Regarding CESM, as you note, the bias computed in this paper is in contrast to previous offline evaluation of CESM's component land model, CLM. While it is possible, or even probable that biases in CESM1 precipitation and air temperature are responsible for the differences

in SCF, I don't believe the Wang et al. study, which is referenced in the text, examines this bias for CESM's atmospheric model. It would be helpful to confirm any air temperature or precipitation biases in CESM1 compared to the corrected MERRA output used by the Toure et al. study to examine CLM. For Figure 2, the range of correlations between SAE anomalies in the NOAA-CDR and the CESM-LE or CMIP5 ensembles are expected since the specific climate trajectories of the historical ensembles are not constrained by observations. Presumably the positive values obtained for the ensemble mean correlations with the NOAA-CDR result from negative trends in the datasets. Also note that NOAA-CRD has been shown to have erroneous trends in SAE during the fall and potentially during the spring as well (Brown and Derksen, Hori et al., Mudryk et al.); it is unclear to what extent this will affect trends in annual-mean SAE.

R: We appreciated above specific comments. 1) For evaluation of snow cover performance in both CMIP5 historical simulations and CESM-LE, we agree that there are plenty of studies (some of them have been listed in the reviewer#1's comments). We used the MODIS and NOAA-CDR data to evaluate the CESM_LE and CMIP5 because those two datasets are best available snow cover data sets over the northern hemisphere. In the following context, we mainly concerned the annual mean of snow cover. Therefore, we did not examine the seasonal biases of the simulations. We agree the conclusion: "snow cover biases change over the course of the season and that annual mean SCF may have lower bias due to compensating differences throughout the year." Actually, if we evaluated any model simulated variables, we can draw similar conclusions for most of them. 2) For CESM simulated snow cover, in particular, for current CESM-LE simulations, there have not any study extensively examined so far. We have made clear that both Toure et al and Wang et al were offline simulations. Wang et al. (2016) used four reanalysis-based atmospheric forcing datasets to offline drive the CLM4.5, and then evaluate the model simulations under different atmospheric variables (in particular, precipitation). The different forcing datasets are somehow alike for the different CESM simulations. We did not intend to evaluate the CESM-LE precipitation in the current study which have been well addressed in the Kay et al. (2015). 3) "the

range of correlations between SAE anomalies in the NOAA-CDR and the CESM-LE or CMIP5 ensembles are expected since the specific climate trajectories of the historical ensembles are not constrained by observations." We agree this point. Unless the data assimilations involved, the earth system model simulations were not constrained by the observations. However, the data assimilation has not been implemented in the earth system model as far as I know.

Figure 3: These results appear consistent with results in IPCC Fifth Assessment Report (e.g. Figure 12.32 in Chapter 12 of the Working Group I contribution). I don't believe the projections of annual mean SAE from the CESM historical ensemble or low warming ensembles are sufficiently unusual to require an update of these results. In particular the historical ensembles appear consistent with the differences in NH LSAT (colder and more snow cover in CESM-LE than CMIP5) and the projections of SAE for each of the four scenarios reflect the various levels of warming.

R: From the documents of CESM-LE or CESM low warming ensemble projects, they did not mention anywhere to update the CMIP5 products. There is also not the issue of current study. The CESM low warming ensemble project is "to achieve long term 1.5 and 2 deg temperature in a stable climate (Sanderson et al. 2017) ", which can be used to assess the climate impacts at different temperate levels, and then to provide some results for the UNFCCC (2015) and IPCC SR1.5 (2018). As we stated in both previous response comments and the manuscript introduction, the four RCP scenarios were designed based on radiation forcing at the top of atmosphere, but not for specific temperature increasing levels. This leads to the timing of 1.5 or 2 deg for different models are different (Jiang et al. 2016 in the reference list). Therefore, CMIP5 model data are not suitable to investigate the climate sensitivity on the specific warming levels, say 1.5 deg or 2deg.

Figure 4: The increase in standard deviation of SAE anomalies reflects the spread in CMIP5 model SAE sensitivity (different models lose snow at different rates during the 20th and 21st centuries, probably mostly due to the spread in warming rates over

the 21st century, although this latter point is not examined by the authors). For the single model ensembles (CESM-LE hist, 1.5deg and 2.0deg) this spread is not being sampled.

R: Thanks for the comment. We will examine relationship of the SAE spread and warming rates, in particular, for CESM_LE hist, 1.5deg and 2.0deg in the revision. And the corresponding results and discussions will be also added in the manusccript.

Figure 5 and 6: These are the first figures which deal with the topic of the paper as discussed in the abstract and introduction. It's unclear why CMIP5 models were examined in the preceding portion of the paper given that they are not used here (although I grant there are potential difficulties in comparing model output at a given warming level taken from transient RCP scenarios and partially equilibrated simulations such as the CESM low warming simulations). While the results presented in Figures 5 and 6 appear reasonable, I don't believe they are analyzed sufficiently or novel enough to form the basis of this paper.

R: Thanks. We used the CMIP5 simulations to compare SCF and temperature evolutions with specific warming levels from CESM low warming project. Since the CESM low warming is the first fully coupled equilibrium climate simulations based on the earth system model, they are deserved to compare with the widely used products of CMIP5. We will add addition discussions in the manuscript.

"The spatial pattern of SNR for both SCF and LSAT are broadly consistent with each other over snow regions, but their magnitude for SCF is much smaller than that for LSAT." It's not clear to me which aspects of the SNR patterns are consistent between that of SCF and that of LSAT.

R: Thanks. We had added more statements to clarify this. For example, over snow regions, we can find that the SNR of both SCF and LSAT are relatively small over Eurasian middle-to-high latitudes compare to other regions, but great over Easter part of USA (east of 90W) and along coaster of Rocket mountains. Over low latitude (south

of 30N, except of Tibetan Plateau) where no snow exists in general, the SNR of LSAT is much great. More detail explanations will be added in the manuscript.

Figure 7: Figure 7b is not providing any additional information from 7a. To the level of detail presented, the results do not appear depend to depend on the forcing scenario. No interpretation is provided or discussed regarding the seasonal dependence.

R: The correlation coefficient of Figure7a does not clearly show the dependence of SFC changes on warming levels or forcing scenarios. Because the correlation only represents the linear relationship of two quantities, the significant negative correlation means the opposite changes of two. The actually dependences of SFC changes on warming levels were presented Figure 5. Figure 7b is to try to quantify the above dependences of SFC change on LSAT change in different seasons. Since the patterns of Figure 7b is pretty similar with Figure 7a, we would delete Figure 7b in the revision, but keep the CD value in the manuscript. Moreover, the discussions about the seasonal dependence of above correlation coefficients would be also added.

Brown, R., and Derksen, C.: Is Eurasian October snow cover extent increasing? Environmental Research Letters, 8, 024006, doi:10.1088/1748-9326/8/2/024006, 2013. Brutel-Vuilmet, C., Ménégoz, M., and Krinner, G.: An analysis of present and future seasonal Northern Hemisphere land snow cover simulated by CMIP5 coupled climate models, The Cryosphere, 7, 67-80, https://doi.org/10.5194/tc-7-67-2013, 2013. Hori, M., Sugiura, K., Kobayashi, K., Aoki, T., Tanikawa, T., Kuchiki, K., Niwano, M., and Enomoto, H.: A 38-year (1978–2015) Northern Hemisphere daily snow cover extent product derived using consistent objective criteria from satellite-borne optical sensors. Remote Sens. Environ., 191, 402–418, Doi:10.1016/j.rse.2017.01.023, 2017. Li, Y., T. Wang, Z. Zeng, S. Peng, X. Lian, and S. Piao (2016), Evaluating biases in simulated land surface albedo from CMIP5 global climate models, J. Geophys. Res. Atmos., 121, 6178–6190, doi:10.1002/2016JD024774. Mudryk, L. R., P. J. Kushner, C. Derksen, and C. Thackeray (2017), Snow cover response to temperature in observational and climate model ensembles, Geophys. Res. Lett., 44, 919–926, doi:10.1002/2016GL071789.

[Figure]

Rupp, D.E., P.W. Mote, N.L. Bindoff, P.A. Stott, and D.A. Robinson, 2013: Detection and Attribution of Observed Changes in Northern Hemisphere Spring Snow Cover. J. Climate, 26, 6904–6914, https://doi.org/10.1175/JCLI-D-12-00563.1 Thackeray, C. W., C. G. Fletcher, and C. Derksen (2015), Quantifying the skill of CMIP5 models in simulating seasonal albedo and snow cover evolution. J. Geophys. Res. Atmos., 120, 5831–5849. doi: 10.1002/2015JD023325. R: Thanks for above references. We will carefully read above the papers, and the relevant papers will be added in the reference list of the manuscript.

---

## Referee Comment (RC2) · Anonymous Referee #2 · 29 Jan 2018

General Comments

This paper explores the changes in snow extent in the Northern Hemisphere projected under 1.5C and 2.0C warming. The snow cover fraction (SCF) for pre-industrial and end of century periods are analysed and compared. Furthermore, the role of surface temperature in diminishing snow cover extent is examined. This paper uses simple comparison techniques, including spatial and temporal correlations and trend analysis to determine the changes in SCF. Whilst these methods are appropriate for this work and the results are robust, the presentation of this paper is significantly lacking refinement and the results do not provide significant new information to the literature. Grammatical mistakes are common and general use of the English language is poor.

Specific Comments

[Figure]

Many of the early figures seem unnecessary, this manuscript could be significantly condensed by removing these figures and making more appropriate use of the literature. For example, Figure 7a and b are repetitive.

In the analysis of variability in section 5:

- Pg 10, line 1-4: do you mean trends or variability – the variability in the period you are talking about does not seem to change, whilst there is a slight decreasing trend. Please check your terminology.

- Is Figure 4 necessary – many studies have shown that the increase spread of the CMIP5 ensembles is due to model spread. A more informative discussion would be around the uncertainties within the CESM-LE model. This also raises the question as to whether or not the CMIP5 models provide any additional value to this manuscript?

Please provide some discussion of the caveats associated with this work and how future work may address these issues, for example, satellite biases, climate model biases.

It would be useful to present the results in a table (like Table 1) and when discussing area change, use percentage change as well to provide meaningful comparison. Some area averages (eg. North America, Europe, Asia) would also help aid this discussion.

You have discussed the role of LSAT on snowfall, and found it to be contributing to between 10-55% of changes in snow cover, depending on seasons. What else could be contributing? Changes in atmospheric circulation, precipitation trends? Please provide some discussion around this.

Technical Comments

There are too many grammatical mistakes/English language problems in this paper to list individually. Some common mistakes are:

SNR stands for Signal to Noise Ratio, CD for coefficient of determination or (R2),

please make sure your terminology is correct

Use of tense – please use either past or present tense consistently throughout the paper

Please review the structure of all sentences. To ensure clarity in writing, put the subject of the sentence at the beginning

Please ensure you know the meaning of the words you are using, and you are using them in the right manner. For example: initialled, annual (I think you mean inter-annual), consensus, reproductions. Please make sure you are using adjectives and adverbs appropriately.

Please ensure all content is relevant, I think this manuscript can be significantly shortened

Please be more specific throughout, for example: Page 2, line 17: Rate of what? Page 3, line 14-15: What scientific gaps? Page 4, line 18: m is a parameter of what? Page 5, line 24: Different evolutions how?

---

## Author Comment (AC2) · 24 Feb 2018

Reply reviewer#1 comments: The authors thank for the reviewer#1 critical comments. According to the ESD interactive discussion process, we will first try to reply the reviewer#1' comments in this short text, and then we will revise our manuscript throughout.

This study purports to examine the response of Northern Hemisphere snow cover in the Community Earth System Model 1 (CESM1) under 1.5_C and 2.0_C of global mean warming above preindustrial levels. I am unable to recommend further publication of this manuscript as it provides a very cursory examination of the question under discussion and in my opinion, does not reach substantial conclusions that have not

been reported elsewhere. The methods used appear reasonable, however the scientific aims and results are poorly communicated throughout the paper. R: The snow simulations were essentially evaluated and reported in many literatures. We appreciated the reviewer# 1 listed a bunch of papers about snow in the models or in the observations. However, none of them discussed the snow cover change under different warming levels, say 1.5 and 2.0deg, which is one of our purposes of current study. Indeed, there are some works who used the CMIP5 outputs to examine the relationship between snow cover and surface air temperature. As we described in the introduction, the CMIP5 simulations under RCP scenarios were the model response from the different Radiation Forcing at the top of the atmosphere, but not for specific surface air temperature warming levels. The CESM low warming project aimed to the specific (i.e., 1.5 and 2 deg) warming targets. Moreover, this manuscript has also discussed the impacts of the climate internal variability and external forcing on the response of snow cover on the different warming levers, which haven't been addressed and reported anywhere previous.

I provide some more specific criticisms below. The rational for providing Figures 1 and 2 is unclear. Presumably it is to evaluate the simulation of climatological snow cover and its variability for the CMIP5 ensemble and the CESM1 model in particular. These evaluations have been performed elsewhere for CMIP5 for a variety of seasons (for example, Li et al. evaluate CMIP5 ensemble mean SCF during DJF (Figure 2b), Thackeray et al. evaluate seasonal evolution of SAE in CMIP5, Brutel-Vuilmet et al. evaluate March/April SCF climatologies in CMIP5, and Rupp et al. evaluate springtime SAE variability from the CMIP5 ensemble). These results suggest that snow cover biases change over the course of the season and that annual mean SCF may have lower bias due to compensating differences throughout the year. Regarding CESM, as you note, the bias computed in this paper is in contrast to previous offline evaluation of CESM's component land model, CLM. While it is possible, or even probable that biases in CESM1 precipitation and air temperature are responsible for the differences in SCF, I don't believe the Wang et al. study, which is referenced in the text, examines

this bias for CESM's atmospheric model. It would be helpful to confirm any air temperature or precipitation biases in CESM1 compared to the corrected MERRA output used by the Toure et al. study to examine CLM. For Figure 2, the range of correlations between SAE anomalies in the NOAA-CDR and the CESM-LE or CMIP5 ensembles are expected since the specific climate trajectories of the historical ensembles are not constrained by observations. Presumably the positive values obtained for the ensemble mean correlations with the NOAA-CDR result from negative trends in the datasets. Also note that NOAA-CRD has been shown to have erroneous trends in SAE during the fall and potentially during the spring as well (Brown and Derksen, Hori et al., Mudryk et al.); it is unclear to what extent this will affect trends in annual-mean SAE.

R: We appreciated above specific comments. 1) For evaluation of snow cover performance in both CMIP5 historical simulations and CESM-LE, we agree that there are plenty of studies (some of them have been listed in the reviewer#1's comments). We used the MODIS and NOAA-CDR data to evaluate the CESM_LE and CMIP5 because those two datasets are best available snow cover data sets over the northern hemisphere. In the following context, we mainly concerned the annual mean of snow cover. Therefore, we did not examine the seasonal biases of the simulations. We agree the conclusion: "snow cover biases change over the course of the season and that annual mean SCF may have lower bias due to compensating differences throughout the year." Actually, if we evaluated any model simulated variables, we can draw similar conclusions for most of them. 2) For CESM simulated snow cover, in particular, for current CESM-LE simulations, there have not any study extensively examined so far. We have made clear that both Toure et al and Wang et al were offline simulations. Wang et al. (2016) used four reanalysis-based atmospheric forcing datasets to offline drive the CLM4.5, and then evaluate the model simulations under different atmospheric variables (in particular, precipitation). The different forcing datasets are somehow alike for the different CESM simulations. Furthermore, in Toure et al offline simulation, the precipitation in atmospheric forcing datasets are from GPCP-bias-corrected MERRA (which are essentially the same as the GPCP on monthly/annual time scale). In the

below figure (Fig. A), we plot the NH land area mean annual precipitation from GPCP, CESM_LE, and it shows that precipitation in CESM_LE is less than that in that in GPCP about 0.059 mm/day for annual mean of 1979-2005. The results have been added in the revision. 3) "the range of correlations between SAE anomalies in the NOAA-CDR and the CESM-LE or CMIP5 ensembles are expected since the specific climate trajectories of the historical ensembles are not constrained by the observations." We agree this point. Unless the data assimilations involved, the earth system model simulations were not constrained by the observations. However, the data assimilation has not been implemented in the earth system model as far as I know.

Figure A Annual NH land average precipitation from CESM-LE (black), CESM-LE ensemble mean (Red) and GPCP (Blue) for period of 1979-2005. The mean value is 2.13 mm/day for CESM-LE ensemble mean (with standard deviation 0.0059mm/day from 40 member), and 2.09 mm/day for GPCP.

Figure 3: These results appear consistent with results in IPCC Fifth Assessment Report (e.g. Figure 12.32 in Chapter 12 of the Working Group I contribution). I don't believe the projections of annual mean SAE from the CESM historical ensemble or low warming ensembles are sufficiently unusual to require an update of these results. In particular the historical ensembles appear consistent with the differences in NH LSAT (colder and more snow cover in CESM-LE than CMIP5) and the projections of SAE for each of the four scenarios reflect the various levels of warming. R: From the documents of CESM-LE or CESM low warming ensemble projects, they did not mention anywhere to update the CMIP5 products. There is also not the issue of current study. The CESM low warming ensemble project is "to achieve long term 1.5 and 2 deg temperature in a stable climate (Sanderson et al. 2017) ", which can be used to assess the climate impacts at different temperate levels, and then to provide some results for the UNFCCC (2015) and IPCC SR1.5 (2018). As we stated in both previous response and the manuscript introduction section, the four RCP scenarios were designed based on radiation forcing at the top of atmosphere, but not for specific temperature increasing

levels. This leads to the timing of 1.5 or 2 deg for different models are different (Jiang et al. 2016 in the reference list). Therefore, CMIP5 model data are not suitable to investigate the climate sensitivity on the specific warming levels, say 1.5 deg or 2deg. We also added discussions to clarify above points.

Figure 4: The increase in standard deviation of SAE anomalies reflects the spread in CMIP5 model SAE sensitivity (different models lose snow at different rates during the 20th and 21st centuries, probably mostly due to the spread in warming rates over the 21st century, although this latter point is not examined by the authors). For the single model ensembles (CESM-LE hist, 1.5deg and 2.0deg) this spread is not being sampled. R: Thanks for the comment. In the supplement materials, Figure S1 shows the standard deviation (spread) of LSAT anomalies which display similar temporal variations as those in Figure 4 (SAE spread). Suggested by the reviewer, we examined relationship of the SAE spread and warming rates, in particular, for CESM_LE hist, 1.5deg and 2.0deg in the revision. To investigate the dependence of SAE change on LSAT, we regressed annual SAE anomaly to LSAT anomaly in each CESM simulation and each CMIP5 model, and then we computed the ensemble mean of regression coefficient and its corresponding standard deviation, respectively. For 2006-2100, the regression coefficient (unit: 106km2/ïĆřC) is -1.37ïĆś0.56 (1.5ïĆřC), -1.12ïĆś0.07(2.0ïĆřC), -1.18ïĆś0.19(RCP2.6), and -0.97ïĆś0.44 (RCP4.5), respectively. For 1920-2005, the regression coefficient is -0.79ïĆś0.42 (CESM-LE), and -0.84ïĆś0.22 (CMIP5). The about values do not show obviously that the dependence of SAE loss on the warming rate in CMIP5 is greater than that from the simulations in CESM. However, based on both Figure4 and Figure S1, we argue that the inter-model diversity of CMIP5 is probably responsible for the increasing in the spread of both SAE and LSAT.

Figure 5 and 6: These are the first figures which deal with the topic of the paper as discussed in the abstract and introduction. It's unclear why CMIP5 models were examined in the preceding portion of the paper given that they are not used here (although I grant there are potential difficulties in comparing model output at a given warming

level taken from transient RCP scenarios and partially equilibrated simulations such as the CESM low warming simulations). While the results presented in Figures 5 and 6 appear reasonable, I don't believe they are analyzed sufficiently or novel enough to form the basis of this paper. R: Thanks. We used the CMIP5 simulations to compare SCF and temperature evolutions with specific warming levels from CESM low warming project. Since the CESM low warming is the first fully coupled equilibrium climate simulations based on the earth system model, they are deserved to compare with the widely used products of CMIP5. Addition discussions have been added in the manuscript.

"The spatial pattern of SNR for both SCF and LSAT are broadly consistent with each other over snow regions, but their magnitude for SCF is much smaller than that for LSAT." It's not clear to me which aspects of the SNR patterns are consistent between that of SCF and that of LSAT. R: We had added more statements to clarify this. For example, over snow regions, we can find that the SNR of both SCF and LSAT are relatively small over Eurasian middle-to-high latitudes compare to other regions, but great over Easter part of USA (east of 90W) and along coaster of Rocket mountains. Over low latitude (south of 30N, except of Tibetan Plateau) where no snow exists in general, the SNR of LSAT is much great. More detail explanations will be added in the manuscript.

Figure 7: Figure 7b is not providing any additional information from 7a. To the level of detail presented, the results do not appear depend to depend on the forcing scenario. No interpretation is provided or discussed regarding the seasonal dependence. R: The correlation coefficient of Figure7a does not clearly show the dependence of SFC changes on warming levels or forcing scenarios. Because the correlation only represents the linear relationship of two quantities, the significant negative correlation means the opposite changes of two. The actual dependences of SFC changes on warming levels were presented Figure 5. Figure 7b trys to quantify the above dependences of SFC change on LSAT change in different seasons. Since the patterns of Figure 7b is similar with Figure 7a, we will delete Figure 7b in the revision, but keep the CD value in

the manuscript. Moreover, the discussions about the seasonal dependence of above correlation coefficients would be also added.

Brown, R., and Derksen, C.: Is Eurasian October snow cover extent increasing? Environmental Research Letters, 8, 024006, doi:10.1088/1748-9326/8/2/024006, 2013. Brutel-Vuilmet, C., Ménégoz, M., and Krinner, G.: An analysis of present and future seasonal Northern Hemisphere land snow cover simulated by CMIP5 coupled climate models, The Cryosphere, 7, 67-80, https://doi.org/10.5194/tc-7-67-2013, 2013. Hori, M., Sugiura, K., Kobayashi, K., Aoki, T., Tanikawa, T., Kuchiki, K., Niwano, M., and Enomoto, H.: A 38-year (1978–2015) Northern Hemisphere daily snow cover extent product derived using consistent objective criteria from satellite-borne optical sensors. Remote Sens. Environ., 191, 402–418, Doi:10.1016/j.rse.2017.01.023, 2017. Li, Y., T. Wang, Z. Zeng, S. Peng, X. Lian, and S. Piao (2016), Evaluating biases in simulated land surface albedo from CMIP5 global climate models, J. Geophys. Res. Atmos., 121, 6178–6190, doi:10.1002/2016JD024774. Mudryk, L. R., P. J. Kushner, C. Derksen, and C. Thackeray (2017), Snow cover response to temperature in observational and climate model ensembles, Geophys. Res. Lett., 44, 919–926, doi:10.1002/2016GL071789. Rupp, D.E., P.W. Mote, N.L. Bindoff, P.A. Stott, and D.A. Robinson, 2013: Detection and Attribution of Observed Changes in Northern Hemisphere Spring Snow Cover. J. Climate, 26, 6904–6914, https://doi.org/10.1175/JCLI-D-12-00563.1 Thackeray, C. W., C. G. Fletcher, and C. Derksen (2015), Quantifying the skill of CMIP5 models in simulating seasonal albedo and snow cover evolution. J. Geophys. Res. Atmos., 120, 5831–5849. doi: 10.1002/2015JD023325. R: Thanks for above references. We will carefully read above the papers, and the relevant papers will be added in the reference list of the manuscript.

Please also note the supplement to this comment:
https://www.earth-syst-dynam-discuss.net/esd-2017-91/esd-2017-91-AC2-supplement.pdf

**Supplement:**

[revised manuscript text omitted]
.  Across four scenarios in annual time scale, the ensemble mean magnitude change of SAE is largest in RCP4.5 (-14.47%), followed by 2.0 °C (-10.92%), RCP2.6 (-8.50%) and the smallest in 1.5°C (-8.02%). For specific scenario in different season, the most SAE loss is in JJA, followed by OSN, and the least is in DJF. The largest percentage loss in JJA is mainly caused by the smallest SAE in 1971-2000, which is the denominator in computation. During 1971-2000, the SAE is 4.56 (3.50) × $10^6$ km$^2$ from CESM-LE (CMIP5) ensemble mean in JJA, while this value is 43.23 (35.99) × $10^6$ km$^2$ in DJF, respectively. Table 2 also shows that the STDs of SAE loss due to ensemble variability is much larger in CMIP5 models than those in the CESM ensemble. For example, the STDs of annul mean percentage is 5.71%, 5.58%, 0.52%, and 0.78% in RCP4.5, RCP2.6, 2.0°C, and 1.5°C scenarios, respectively. This further implies the inter-model variability will induce relatively greater SAE uncertainty than those due to internal climate variability.

[revised manuscript text omitted]

---

## Author Comment (AC3) · 24 Feb 2018

General Comments This paper explores the changes in snow extent in the Northern Hemisphere projected under 1.5C and 2.0C warming. The snow cover fraction (SCF) for pre-industrial and end of century periods are analysed and compared. Furthermore, the role of surface temperature in diminishing snow cover extent is examined. This paper uses simple comparison techniques, including spatial and temporal correlations and trend analysis to determine the changes in SCF. Whilst these methods are appropriate for this work and the results are robust, the presentation of this paper is significantly lacking refinement and the results do not provide significant new information

to the literature. Grammatical mistakes are common and general use of the English language is poor. R: We are thankful the reviewers' comments and suggestions. We have made efforts to improve the manuscript. The English writing has been edited. We have also made additional analyses, and the corresponding texts have been added in the manuscript.

Specific Comments Many of the early figures seem unnecessary, this manuscript could be significantly condensed by removing these figures and making more appropriate use of the literature. For example, Figure 7a and b are repetitive. R: Thanks. Figure 7b are removed.

In the analysis of variability in section 5: - Pg 10, line 1-4: do you mean trends or variability – the variability in the period you are talking about does not seem to change, whilst there is a slight decreasing trend. Please check your terminology. R: Thanks. We have corrected the misunderstanding expression. The variation can be represented by the standard deviation. Here we mean the linear trend.

- Is Figure 4 necessary – many studies have shown that the increase spread of the CMIP5 ensembles is due to model spread. A more informative discussion would be around the uncertainties within the CESM-LE model. This also raises the question as to whether or not the CMIP5 models provide any additional value to this manuscript? R: Indeed, there have been discussed in many studies, which focused on the precipitation and temperature, and their derivation. We persist remaining Figure 4 and the reasons are: 1) in this manuscript, we discuss about land snow cover area (SCE) which has not been extensively investigated under different warming scenarios, in particular, CESM-low warming scenarios. 2) those curves give the temporal variations of SCE spread due to inter-model spread or internal climate variability from historical period till 2100, and 3) because we analyze the snow cover from multi models, it is necessary to quantify the spread of the ensemble. In the revision, we have added texts to emphasis above reasons. To investigate the dependence of SAE change on LSAT, we regressed annual SAE anomaly to LSAT anomaly in each CESM simulation and each

CMIP5 model, and then we computed the ensemble mean of regression coefficient and its corresponding standard deviation, respectively. For 2006-2100, the regression coefficient (unit: 106km2/ïĆřC) is -1.37ïĆś0.56 (1.5ïĆřC), -1.12ïĆś0.07 (2.0ïĆřC), -1.18ïĆś0.19(RCP2.6), and -0.97ïĆś0.44 (RCP4.5), respectively. For 1920-2005, the regression coefficient is -0.79ïĆś0.42 (CESM-LE), and -0.84ïĆś0.22 (CMIP5). The about values do not show obviously that the dependence of SAE loss on the warming rate in CMIP5 is greater than that from the simulations in CESM. However, based on both Figure4 and Figure S1, we argue that the inter-model diversity of CMIP5 is probably responsible for the increasing in the spread of both SAE and LSAT.

Please provide some discussion of the caveats associated with this work and how future work may address these issues, for example, satellite biases, climate model biases. R: Added.

It would be useful to present the results in a table (like Table 1) and when discussing area change, use percentage change as well to provide meaningful comparison. Some area averages (eg. North America, Europe, Asia) would also help aid this discussion. R: Thanks. We computed the SAE changes between 2071-2010 and 1971-2000 in annual and four seasons from four scenarios. A table (Table2) is added to present about above results.

You have discussed the role of LSAT on snowfall, and found it to be contributing to between 10-55% of changes in snow cover, depending on seasons. What else could be contributing? Changes in atmospheric circulation, precipitation trends? Please provide some discussion around this. R: We have added the texts to explain possible contributors on the seasonal variation of snow cover.

Technical Comments There are too many grammatical mistakes/English language problems in this paper to list individually. Some common mistakes are: SNR stands for Signal to Noise Ratio, CD for coefficient of determination or (R2), please make sure your terminology is correct R: Corrected.

Use of tense – please use either past or present tense consistently throughout the Paper R: Done.

Please review the structure of all sentences. To ensure clarity in writing, put the subject of the sentence at the beginning R: Done.

Please ensure you know the meaning of the words you are using, and you are using them in the right manner. For example: initialled, annual (I think you mean inter-annual),consensus, reproductions. Please make sure you are using adjectives and adverbs appropriately. R: Corrected.

Please ensure all content is relevant, I think this manuscript can be significantly shortened R: Thanks. We have deleted some unnecessary texts.

Please be more specific throughout, for example: Page 2, line 17: Rate of what? Page 3, line 14-15: What scientific gaps? Page 4, line 18: m is a parameter of what? Page 5, line 24: Different evolutions how? R: Done.

Please also note the supplement to this comment:
https://www.earth-syst-dynam-discuss.net/esd-2017-91/esd-2017-91-AC3-supplement.pdf

**Supplement:**

[revised manuscript text omitted]
.  Across four scenarios in annual time scale, the ensemble mean magnitude change of SAE is largest in RCP4.5 (-14.47%), followed by 2.0 °C (-10.92%), RCP2.6 (-8.50%) and the smallest in 1.5°C (-8.02%). For specific scenario in different season, the most SAE loss is in JJA, followed by OSN, and the least is in DJF. The largest percentage loss in JJA is mainly caused by the smallest SAE in 1971-2000, which is the denominator in computation. During 1971-2000, the SAE is 4.56 (3.50) × $10^6$ km$^2$ from CESM-LE (CMIP5) ensemble mean in JJA, while this value is 43.23 (35.99) × $10^6$ km$^2$ in DJF, respectively. Table 2 also shows that the STDs of SAE loss due to ensemble variability is much larger in CMIP5 models than those in the CESM ensemble. For example, the STDs of annul mean percentage is 5.71%, 5.58%, 0.52%, and 0.78% in RCP4.5, RCP2.6, 2.0°C, and 1.5°C scenarios, respectively. This further implies the inter-model variability will induce relatively greater SAE uncertainty than those due to internal climate variability.

[revised manuscript text omitted]

---

## Author Response (AR2)

**Point-to-point to reply editors and reviewers' comments**

*Editor Decision: Reconsider after major revisions*

*We have sent this manuscript to review by orginal reviewers in the last around of review and an additional reviewer. Based on the comments, I decide to send back to the authors for major revisions on this manuscript. As pointed by referee #2, the authors should improve the English of this manuscript substantially by an English native speaker before the manuscript can be considered to publish in the ESD. Suggestions for revision or reasons for rejection (will be published if the paper is accepted for final publication)*

*I have examined the other reviewers' comments on the earlier version of the manuscript and authors' point-by-point replies. I find that the authors' replies are generally adequate in addressing the relevant concerns that the reviewers raised. I recommend acceptance of the manuscript for publication after careful editing and checking figure captions.*

R: In this revision, we have consigned a Professional Proofreading and Editing Service from the United States to proofread the English writing. We also carefully read and edit the possible writing errors. Both the revision-with-track-change and the revision-with-changes-incorporated are attached.

*Reviewer #1*

*I have examined the other reviewers' comments on the earlier version of the manuscript and authors' point-by-point replies. I find that the authors' replies are generally adequate in addressing the relevant concerns that the reviewers raised. I recommend acceptance of the manuscript for publication after careful editing and checking figure captions.*

R: We appreciate the reviewer#1 comments and suggestions. We have careful checked and corrected all figures' caption.

2nd Reviews

*The authors have taken into consideration the specific comments of my review and have addressed them satisfactorily. The results presented are robust and provide a useful analysis of snow cover change now and into the future.*

*However, the technical comments surrounding the use of the English language have not been addressed. Whilst I can see that the manuscript has been thoroughly revised, the numerous grammatical problems remain and more have been introduced. I cannot recommend that this manuscript be published until these issues have been resolved. This will mean a proof read by someone other than the Authors, I would recommend either a proof read by a native English speaker or preferably a paid proof reading service.*

R: We appreciate the reviewer#2 comments and suggestions. We have consigned a Professional Proofreading and Editing Service from the United States to proofread the English writing. We also carefully read and edit the possible writing errors.

*Minor Comments:*

*Page 8, Line 4: STD should be standard deviation not derivation*

R: Corrected.

*Page 14, Line 2: Coefficient of determination, not correlation of determination*

R: Corrected

*Throughout: The mountain ranges in North America are called the Rocky Mountains, not the Rocket Mountains.*

R: Corrected.

*Check that your acronyms are only defined once*

R: Checked.

[revised manuscript text omitted]